# Novel Deep Sea Isoindole Alkaloid FGFC1 Exhibits Its Fibrinolytic Effects by Inhibiting Thrombin-Activatable Fibrinolysis Inhibitor

**DOI:** 10.3390/ph17101401

**Published:** 2024-10-20

**Authors:** Haixing Zhang, Xiaozhen Diao, Tingting Jiang, Mingjun Wei, Yue Su, Jingjing Shen, Chunlin Bao, Wenhui Wu

**Affiliations:** 1Department of Marine Bio-Pharmacology, College of Food Science and Technology, Shanghai Ocean University, Shanghai 201306, China; starfishhx@163.com (H.Z.); xzdiao@shou.edu.cn (X.D.); j15037793607@163.com (T.J.); weimj98@163.com (M.W.); suyue0616@163.com (Y.S.); 2Putuo Sub-Center of International Joint Research Center for Marine Biological Sciences, Zhoushan 316000, China; 3Analytical & Measuring Instruments Division Shimadzu (China) Co., Ltd., Shanghai Branch, Shanghai 200120, China; sshshjj@shimadzu.com.cn; 4Shanghai Sixth People’s Hospital Affiliated to Shanghai Jiao Tong University School of Medicine, Shanghai 201306, China; 5Marine Biomedical Science and Technology Innovation Platform of Lin-Gang Special Area, Shanghai 201306, China

**Keywords:** thrombin-activated fibrinolytic inhibitor, deep sea bioactive compound, coagulation, fibrinolysis, thrombomodulin

## Abstract

Background: The thrombin-activatable fibrinolysis inhibitor (TAFI) is an important regulator in the balance between blood clot formation (coagulation) and dissolution (fibrinolysis), which is mainly activated by thrombin bonded with thrombomodulin (TM). Methods: In this study, the investigation focused on the unique target TAFI of fungi fibrinolytic compound 1 (FGFC1), a novel fibrinolytic compound sourced from the deep sea. In this sense, the regulation of TAFI by FGFC1, in comparison to established TAFI inhibitors such as DS-1040 and PCTI in hPPP, was investigated, which was validated through the molecular docking of FGFC1 to TAFI. The inhibitory effect of FGFC1 on TAFI-mediating coagulation (ex vivo and in vitro) and its fibrinolytic effect (ex vivo) were investigated in hPPP and hCMEC/D3 cells, respectively, followed by SEM. Results: FGFC1 solutions ranging from 0.023 to 0.736 mM effectively inhibited TAFI activation. Notably, the 0.023 mM concentration demonstrated significant suppression, comparable to DS-1040 and PCTI. These inhibitory effects of FGFC1 (0.023–0.368 mM) were further validated through the enhancement in TAFI (TAFIa) activation by fibrins in the coagulum prior to proteolysis, resulting in the cleavage of TAFIa from 33 kDa to 28 kDa. Furthermore, these regulatory effects of FGFC1 on TAFI were demonstrated to have minimal association with TM-mediated control, as confirmed through a molecular docking analysis. FGFC1 (0.023–0.092 mM) was suggested to have obstructive effects on TAFI-mediated coagulation in the hPPP, which was demonstrated by the inhibition of clot aggregation, protein crystallization, and platelet anchoring, as observed through SEM. Simultaneously, FGFC1 (0.023 to 0.368 mM) significantly enhanced TAFI-mediated fibrinolysis, which was also supported by increased levels of t-PA, u-PA, and plasmin. Conclusions: From the above findings, FGFC1 is identified as a novel dual-target bioactive compound participating in blood formation/dissolution that demonstrates anti-coagulation and fibrinolytic effects by regulating TAFI activation, inhibiting TAFIa–fibrin combination, and initiating proteolysis. It also provided convincing evidence that TAFI plays a critical role in thrombolysis as a molecular link between coagulation and fibrinolysis. Furthermore, the application of FGFC1 was indicated as a potential therapeutic strategy in thromboembolic and hemorrhagic diseases.

## 1. Introduction

Thromboembolic diseases, including coronary heart disease, hemorrhagic stroke, and venous thromboembolism, are the most common diseases threatening human life and health [1,2,3]. So far, there have been three main kinds of drugs for the treatment of these diseases depending on thrombosis composition, including fibrinogen clots and aggregates of platelets: platelet aggregation inhibitors [4,5], anti-coagulants [6,7], and thrombolytic drugs [8,9]. Unfortunately, most of these thrombolytic drugs show either unavoidable side effects (potential bleeding complications, allergy, and liver function damage) or poor therapeutic efficacy (narrow therapeutic window and short half-life), which calls for their further development. In cardiovascular disease studies, TAFI, which was discovered by Hendriks et al. in serum in 1989 and previously defined as carboxypeptidase U (CPU), deserves equal attention among researchers typically focusing on blood clotting or the fibrinolytic system due to its important role in maintaining blood circulation and the delicate balance between coagulation and fibrinolysis [10,11,12,13]. After blood clots caused by vascular damage occur, TM-mediated TAFI activation by thrombin prevents fibrinolysis from occurring prematurely, which suppresses fibrinolytic activation to guarantee coagulation against persistent bleeding [14,15,16,17]. Depending on its partial elimination of Arg, which is meant to be recognized by fibrinolysin, two pathways, namely interference with TAFI activation and the direct inhibition of TAFI, are indicated as promising pharmacological strategies for the treatment of thrombosis and cardiovascular diseases [18,19,20,21]. These pathways could be investigated by the unique properties of TAFI in relation to coagulation morphology and the fibrinolytic system [22].

According to our previous research, as a novel rare marine isoindole alkaloid, FGFC1 (C_51_H_68_N_2_O_10_, 869.1g/mol, PubChem CID:102219046, Figure 1), which is one of the secondary metabolites produced by the *Stachybotrys longispora* strain FG216, has been proven to have a remarkable fibrinolytic activity (fibrin was found to be completely dissolved by 10 mg/kg FGFC1 in rats) with low toxicity (EC_50_ as 5 μM) [23,24,25,26,27,28,29]. Although relatively comprehensive investigations on the mechanism of the fibrinolytic effect of FGFC1 have been conducted before, the further exploration of TAFI-mediating FGFC1’s thrombolytic effects is still needed.

In this study, an hPPP-based ex vivo evaluating system was established to investigate the TAFI-mediating thrombolytic effect of FGFC1. Compared with previous experiments, the evaluating system in this study adopts a more realistic method that relies on the hPPP to simulate the in vivo blood environment, so it can provide more accurate physiological conditions for the production, activation, and function of TAFI. Based on the system, this study fills out the mechanism of the TAFI-mediating thrombolytic effect of FGFC1, regarding the regulation on the coagulation and fibrinolysis. Furthermore, it would provide theoretical basis for the potential targets of therapeutic strategies related to thromboembolic and hemorrhagic diseases and also inspire the development of thrombolytic drugs.

## 2. Results

### 2.1. Regulatory Effects on TAFI by FGFC1

#### 2.1.1. FGFC1 Inhibited TAFI Activation

The hPPP applied in this study was investigated for the exhibited TAFIa activity compared to that in the human plasma. It was suggested that the sufficient TAFIa activity detected in the hPPP (Abs_405_ = 0.0526 ± 0.0016) was no different from that in plasma (Abs_405_ = 0.0502 ± 0.0021), which was consistent with TAFI’s dominant existence in plasma compared to blood cells [22]. Both TAFI inhibitors including DS-1040 (0.005 mM) and PCTI (0.054 mM), along with FGFC1 (0.023–0.736 mM), inhibited TAFI activation, indicated by the decreasing TAFIa activity detected in the hPPP (Figure 2). The detections of TAFIa activity in the FGFC1 groups were dose-dependent, among which the 0.023 mM group (88.97%) exhibited considerable TAFIa activity compared to DS-1040 (81.07%) and PCTI (86.42%) groups.

#### 2.1.2. FGFC1 Inhibited TAFIa–Fibrin Combination and Initiated Its Proteolysis

In contrast to the Blank group (0.389 Abs), TAFIa activities in the FGFC1 groups (0.023–0.368 mM) were predominantly detected in the solution phase (about 0.45–0.48 Abs), which were much higher than that inside the coagulum (<0.08 Abs). These preferential existences of TAFIa in the specific zone were proven to show a dose-dependent effect in the low concentrations of FGFC1 solutions (0.023–0.368 mM), with a peak level of 0.484 Abs in the 0.368 mM group. However, these preferential existences gradually declined with higher concentrations of FGFC1 solutions (0.553–0.736 mM) and fell to 0.454 Abs in the 0.736 mM group (Figure 3(A1,A2)).

Two main monomers of TAFIa with different molecular weights including 33 kDa and 28 kDa were detected in the solution phase in a valid concentration interval; the former was recognized as the common form of TAFIa after the removal of the activation peptide (~92 AA) from TAFI. Different from the dominant existence of 33 kDa compared to 28 kDa (20:9) detected in the Blank group, the FGFC1 groups showed a much higher level of 28 kDa with no significant difference among the concentrations (Figure 3B).

### 2.2. Inhibitory Effect of FGFC1 on TAFI-Mediating Coagulation Ex Vivo and In Vitro

#### 2.2.1. FGFC1 Inhibited TAFI-Mediating Coagulation in the hPPP

Both the size and weight of the coagulum were reduced by 70–86% and 60–85%, respectively, by FGFC1 (0.023–0.736 mM) (Figure 4A), among which the 0.023 and 0.368 mM groups showed nearly the same inhibitory effects on the coagulation in the hPPP (reduced by about 85%) (Figure 4B). However, high concentrations of the FGFC1 solutions (0.552–0.736 mM) showed a relatively lower inhibitory effect.

#### 2.2.2. FGFC1 Exhibited Little Influence on TM in the hPPP

FGFC1 (0.023–0.736 mM) barely showed any influence on the soluble-TM levels in the hPPP (Figure 5A), which was consistent with that in normal human plasma previously reported. However, FGFC1 solutions (50–175 μM) weakly inhibited the soluble-TM levels in hCMEC/D3 cells in a dose-dependent way (Figure 5B). Therefore, FGFC1’s regulatory effect on TAFI was indicated as a direct impact rather than a consequential influence by its regulation on TM, which further anchored FGFC1’s essential role regarding TAFI.

### 2.3. FGFC1 Reduced Coagulum Level and Obstacled Platelet Anchoring in hPPP

The coagulation level was reflected by the crystallization degree of the protein in the coagulum. No crystallization occurred with the low concentrations of FGFC1 solutions (0.023–0.092 mM). A single crystal started to appear in the 0.184 mM group and increased with the higher concentration, whose number reached the maximum in the 0.736 mM group. The Blank group exhibited a much larger crystallization degree shown by protein crystals at larger dimensions, whose crystal size was 133% of that in the 0.736 mM group (Figure 6A).

The platelet analogs (PAs) on the surface of the coagulum could also be observed, which indicated the coagulation level to some degree. The PAs in the Blank group were observed as black protrusions surrounded by illuminating halos. Small pits indicating the positions where platelets were shed from the surface of the coagulum were observed in the groups with low concentrations of FGFC1 solutions (0.023–0.092 mM). Although PAs could barely be seen in the high-concentration groups (0.368–0.736 mM), much less small pits were shown (Figure 6B).

### 2.4. FGFC1 Facilitated TAFI-Mediating Fibrinolysis Ex Vivo

The positive effects on the uPA and plasmin activities in the hPPP were shown with the FGFC1 solutions (0.023–0.552 mM) and gradually diminished over time within 3 h (Figure 7A,E). The positive effects on the tPA activity were shown with the FGFC1 solutions (0.023–0.092 mM) with the same time-diminishing trend (Figure 7C). Among the effective concentration interval, 0.023 mM FGFC1 contributed to the highest relative activity of uPA (119%), tPA (105%), and plasmin (115%), compared to that in the Blank group (0 mM), after a 150 min reaction since the coagulum formed (Figure 7B,D,F). Contrarily, the high concentration of the FGFC1 solution (0.736 mM) led to the inhibited activities of uPA, tPA, and plasmin in the hPPP, which were even lower than that in the Blank group.

### 2.5. Molecular Docking of FGFC1 to TAFI

Considering the binding energy, the FGFC1-TAFI conformation and FGFC1-TM conformation (−5.62 and −0.52 kcal/mol, respectively) were selected among the top twenty molecular docking conformations (Table 1), in order to further investigate the specificity and selectivity of the ligand conformations. It was suggested that FGFC1 effectively concealed the TAFI active center composed of Zn^2+^ binding sites connecting to SER291 and HIS293 with close hydrogen bonds in TAFI’s catalytic domain (Figure 8A,B). In the FGFC1-TM conformation, FGFC1 was indicated to have hydrogen bonding with ASP60E in the TM–thrombin complex (Figure 8C,D). Compared to the low binding energy between FGFC1 and TM, the FGFC1-TAFI complex showed higher binding energy with TM, which indicates FGFC1’s preferential binding with TAFI rather than directly binding with TM (Figure 8E).

## 3. Discussion

So far, there are two main endogenous anti-fibrinolysis inhibitors highly expressed in the fibrinolysis system, including TAFI and plasma plasminogen activator inhibitor-1 (PAI-1). Different from that TAFI is widely expressed in the plasma, PAI-I is only secreted and expressed in platelets. In this study, the thrombolysis-evaluating ex vivo system constructed by the hPPP with platelets mostly removed was applied to investigate the mechanism of the thrombolytic effect by targeting specific protein, the same for TAFI-mediating procoagulant and anti-fibrinolytic effects. Moreover, similar experiments in platelet-enriched plasma (PRP), plasma, and whole blood should be conducted to investigate FGFC1’s effect on PAI-I or other blood factors.

According to its unique functional properties, the expression level of TAFI is likely to be genetically controlled. This study provided a blood environment closer to the human body for TAFI production, activation, and function, via using the described hPPP [30]. TAFI is synthesized as a propeptide (432 AA), consisting of a 22 AA signal peptide, a 92 AA activation peptide, and a 309 AA catalytic domain [31]. With the removal of the heavily glycosylated activation peptide at Arg92, TAFI would be converted into TAFIa of a 35 kDa molecular weight [32]. With the supplements by FGFC1 solutions, after a 15 min activation by thrombin, TAFIa failed to combine with fibrins at the coagulum before its proteolysis to lose activity, supported by reference [33]. Combining the relatively higher intensity of 28 kDa (compared to 35 kDa) in the FGFC1 solution groups, the remaining TAFI activity in them indicated that TAFIa monomers with a 28 kDa molecular weight still possess the activity. In this case, the comparison on the activity between the two monomers requires further investigation due to the fact that the anti-fibrinolytic function of TAFI mostly depends on its concentration and activation rate and the half-life of TAFIa in plasma [34]. Meanwhile, whether the TAFIa-binding site in fibrin was occupied or interfered by FGFC1 remains to be investigated.

Based on our previous research, FGFC1 at the dose of 5 mg/kg (equivalent to 5.76 μM) showed a comparable fibrinolytic effect to pro-uPA at the dose of 2.7 mg/kg in a rat model of acute pulmonary thrombosis [27]. Additionally, FGFC1 was proven to enhance the fibrinolytic activity mediated by either plasminogen or scu-PA, and to significantly reduce the time required to reach the peak of drug efficacy [28]. The binding of plasminogen to lysine residues on fibrin enhances the efficiency of plasmin activation by tPA and uPA. In human blood, TAFI conditionally interacts with plasminogen to form the complex; it can also recognize and cleave lysine residues on fibrin, thereby inhibiting the formation of plasmin [19]. This study found that FGFC1 exhibited the regulatory effect of uPA and tPA in a dose-dependent manner in the hPPP. The irregular changes in the plasmin activity were shown by various concentrations of FGFC1 supplements, which may indicate that the FGFC1-TAFI interaction interferes with the plasmin activity, and synergizes with FGFC1 to enhance plasmin activity, to provide the fundamental effect for the profibrinolysis (Figure 9).

According to the concentration-dependent dual effect of TM in maintaining the dynamic balance between clot formation and degradation reported, TM at a low concentration enhances TAFI activation by thrombin [35], while facilitating protein C (PC) activation at a high concentration to conversely further act as an obstacle to the coagulation [36]. Although TM is expressed at comparable levels throughout the vasculature, its activation on TAFI outflanks that on PC in the plasma due to the endothelial cell PC receptor necessary for PC activation abundantly expressed on the endothelial cell surface [37]. In this study, TM as the potential mediator in the regulation of FGFC1 on TAFI was investigated in the hPPP, whose levels were further compared with that in hCMEC/D3 cells. Little influence of FGFC1 supplements on TM was observed in PPP, along with weak inhibition on soluble-TM levels only at the high concentration (175 μM) in hCMEC/D3 cells, which may be interpreted as the influence by cell proliferation. Furthermore, the higher binding energy of FGFC1 with TAFI, more remarkable than TM, supported the FGFC1 directly targeting TAFI instead of through TM mediation in PPP. Considering the inhibitory effect by FGFC1 on soluble-TM levels in hCMEC/D3 cells, further investigations focusing on the TM-mediating influence by FGFC1 on TAFI or PC in different endothelial cells, referring to the previous reports [37], should be conducted in the near future.

Hematologic malignancy closely correlated with thrombosis. For instance, the incidence of arterial and venous thrombosis increase when multi-agent chemotherapy and anti-myeloma immunomodulatory drugs are applied in hematologic malignancy, such as multiple myeloma [38]. Also, polycythemia vera, one of the chronic hematologic malignancies, frequently induced thrombosis, hemorrhage, and acute myeloid leukemia [39]. In this case, FGFC1 with anti-coagulation and fibrinolytic effects was indicated as the potential therapeutic strategy in hematologic malignancy inducing thromboembolic and hemorrhagic diseases.

This study aims to investigate the various physiological and pathological processes involved in the formation and dissolution of blood clots (coagulation and fibrinolysis) in relation to thrombosis. The role of TAFI in these processes was a key focus of this research. According to the optimal effective concentration interval regarding different detecting indicators involving regulatory effects on TAFI (0.023–0.368 mM), anti-coagulation (0.023–0.092 mM), and the fibrinolytic effect (0.023–0.552 mM), it may provide a reasonable clinical administration regimen. For instance, administering a small amount of FGFC1 can help with preventing the bleeding side effects during the initial phase of thrombosis, while still achieving the necessary concentration for effective anti-coagulation. In the later stage, the dosage could be appropriately increased to preferentially meet the required concentration for its inhibitory effect on TAFI (or TAFIa), which would definitely fit the wider effective concentration interval of the fibrinolytic effect.

Combining our previous study on the FGFC1’s protective effect on the blood–brain barrier from an injury [40], FGFC1 was indicated as a multi-bioactive compound, potentially participating and exhibiting its positive effect in various steps from the clot formation to the thrombolysis followed by the endothelial barrier injury caused by excessive anti-coagulation. If the regulatory effects by FGFC1 on TAFI found in this study could be dynamically regulated, in this case, TAFI’s preferential mediation of coagulation or fibrinolysis could be controlled during the continuous pathological process. Therefore, it would be promising for FGFC1 to be applied to patients who are pregnant or with bleeding risk.

## 4. Materials and Methods

### 4.1. The Construction of the hPPP-Based Ex Vivo System

The hPPP employed in this study was isolated by the centrifugation (3500 rpm, 4 °C, 10 min) of freshly collected whole blood from healthy volunteers in order to be separated with blood cells and most of the platelets. The hPPP layer was subsequently filtered and washed to generate the PPP samples. The isolation procedure was performed under sterile conditions at sub-zero temperatures. The hPPP acquisition experiments were conducted by the East Branch of Shanghai Sixth People’s Hospital. The hPPP samples were added to a procoagulant (thrombin), plasmin inhibitor (Trypsin Inhibitor), thrombin inhibitor (PPACK), or anti-coagulant (ACD or EDTA), according to the corresponding detected indicators for investigating the regulations on TAFI, anti-coagulation, and the fibrinolytic effect of FGFC1, followed by incubation at 37 °C for 5 min.

### 4.2. Investigation of FGFC1’s Regulatory Effects on TAFI

The 300 μL hPPP samples were separately incubated with different concentrations of the FGFC1 supplement at 37 °C for 5 min to allow the complete reaction between FGFC1 and the PPP. The coagulation in the hPPP was initiated by 500 U/mL thrombin (Macklin, Shanghai, China) at 37 °C for 10 min and was terminated by 150 μM PPACK (Glpbio, Shanghai, China), whose fibrinolysis was inhibited by 2 μM Trypsin Inhibitor (Macklin, Shanghai, China). The formed coagulum and supernatant were separated by centrifugation (10,000 rpm, RT, 10 min) for a further analysis of TAFI activity. The coagulum isolated was re-dissolved in a 1 M Tris-HCl buffer through sonication (25 kHz), and Hippuryl-Arg was utilized as the substrate for assessing TAFI activity. TCT-1,4-Dioxane was prepared by 3.125% Cyanuric chloride (Macklin, Shanghai, China) (*w*/*v*, 1,4-dioxane (Glpbio, Shanghai, China)) for the detection of Arg, whose concentration was measured at 405 nm [41,42].

The molecular weight changes in TAFIa by FGFC1 during the coagulation were analyzed using Matrix-Assisted Laser Desorption Ionization Mass Spectrometry (MALDI-MS). The samples were diluted 20-fold with water, and the MALDI matrix was prepared using a 20 mg/mL solution of 3,5-dimethoxy-4-hydroxycinnamic acid in 50% (*v*/*v*) spectrometry/0.1% (*v*/*v*) trifluoroacetic acid. A calibration reference was prepared using a 500 fmol/μL solution of BSA. Subsequently, 1 μL of the sample was carefully applied onto a stainless steel MALDI sample plate, followed by overlaying with 1 μL of the matrix solution. The MALDI-8030 mass spectrometer (Shimadzu, Kyoto, Japan) was employed for conducting an MALDI in-source decay analysis. The acquired MALDI spectra were processed using the MALDI Solutions Data Acquisition software v2.5.1. The acquisition parameters of MALDI-MS analyses are summarized in Table 2.

### 4.3. ELISA Assays for TM Detection

The hPPP samples were under the anti-coagulated treatment (RT) using EDTA in the Tris-HCL buffer as the anti-coagulant (1.4 mg/mL, pH 7.4) to exclude the differences in absorbance values between solid and liquid phases in the samples. The sample supernatants to be tested were isolated from the precipitate after centrifugation (3000 rpm, RT, 30 min), followed by the incubation with the FGFC1 solutions (50, 80, 100, 120, 150, and 175 μM) at 37 °C for 15 min. The concentration of TM in the supernatants was quantified using a TM ELISA detection kit (MEIMIAN, Shanghai, China).

hCMEC/D3 cells (Meisen, Hangzhou, Zhejiang, China) were used to investigate the effect of FGFC1 on TM on vascular endothelial cells. hCMEC/D3 cells seeded into a 96-well microplate (passage 9–10, 1 × 10^4^ cells/well) were cultured by an ECM medium (Meisen, Hangzhou, Zhejiang, China) in an incubator (5% CO_2_ at 37 °C) until the confluence, followed by a 4 h incubation with FGFC1 solutions (0, 50, 80, 100, 120, 150, and 175 μM). Treated hCMEC/D3 cells were then under lysis by an RIPA Lysis Buffer (Beyotime, Shanghai, China) with PMSF (Beyotime, Shanghai, China) (100:1), followed by centrifugation (12,000 rpm, 4 °C, 10 min) to collect the supernatants to be tested by the TM ELISA detection kit (MEIMIAN, Shanghai, China).

### 4.4. Investigation on the Anti-Coagulation Effect of FGFC1

The coagulum obtained by previously described procedures was pre-frozen at −80 °C for 4 h, followed by a 48 h lyophilization using a vacuum freeze dryer (Labconco Freezone 2.5 L, Kansas, MO, USA). The lyophilized samples were directly affixed onto the conductive glue, and then were coated with a gold layer by utilizing an OxfordQuorumSC7620 (Quorum Technologies, Sacramento, CA, USA) sputter-coating instrument (10 mA, 45 s). The morphologic characteristics of the fixed samples were observed using a ZEISS Sigma300 SEM (ZEISS, Jena, Germany), with an acceleration voltage of 3 kV employed during scanning and imaging.

### 4.5. Investigation on the Fibrinolytic Effect of FGFC1

The hPPP samples were under the anti-coagulated treatment (RT) using ACD anti-coagulants (0.48% citric acid, 1.32% sodium citrate, and 1.47% glucose, *w*/*v*, ddwater). The sample supernatants to be tested were isolated from the precipitate after centrifugation (2000 rpm, RT, 20 min), followed by incubation with various concentrations of FGFC1 solutions (0, 0.023, 0.046, 0.092, 0.184, 0.368, 0.552, and 0.736 mM) at 37 °C for 15 min.

The chromogenic substrates S-2288 [43], S-2444 [44], and S-2551 [45] are highly effective as substrates for tPA, uPA, and plasmin, respectively. The 70 μL sample supernatants were mixed with a 50 μL chromogenic substrate and 30 μL Tris-HCL (0.05 mM, pH 7.4) in a 96-well microplate. The final concentrations of the chromogenic substrates were 1.4 mM for S-2288, 1.8 mM for S-2444, and 1.5 mM for S-2551, respectively. The expressions of tPA, uPA, and plasmin were shown by fluorescence intensity at 405 nm, measured via a microplate reader at 37 °C. The whole testing lasted for 180 min with the measurements conducted every 5 min.

### 4.6. Molecular Docking

Following the experimental approach by Gao et al. [27]., AutoDock tools 1.5.6 was used to simulate the docking of FGFC1 and TAFI. The three-dimensional structure of TAFI protein was obtained from the Protein Data Bank (PDB) (accession number PDB pdb3D68). FGFC1 was visualized and edited using ChemOffice Professional 19, and subsequently converted into 3D structure files by using Open Babel GUI-2.4.1 software (v2.4.1) [46]. The molecular docking procedures were conducted as described in Figure 10.

### 4.7. Statistical Analysis

All experiments were conducted in parallel 6 times. Data were analyzed with GraphPad Prism 8 using the *t*-test. Data were expressed as the mean ± standard deviation (SD) of 3 separate experiments. Two-sided *p* values less than 0.001 were regarded as statistically extremely significant (**** p* < 0.001). All values were indicated as the mean ± SD (n = 6). Statistical significance was determined by the Mann–Whitney U test for the comparison between two groups and the Kruskal–Wallis test for that among groups.

## 5. Conclusions

Through the hPPP-based ex vivo evaluation system utilized in this research, it was demonstrated that deep sea-sourced novel fibrinolytic compound 1 (FGFC1) from marine fungi *S. longispora* strain FG216 could regulate TAFI by inhibiting TAFI activation, preventing TAFIa from binding to fibrin, and initiating the degradation of TAFIa. The regulatory impact of FGFC1 was also shown to play a role in its anti-coagulant properties in relation to clot formation, protein solidification, and platelet adhesion, as well as its fibrinolytic effect involving tPA, uPA, and plasmin. This research addressed the gap in TAFI’s involvement in the anti-coagulant and fibrinolytic mechanisms of FGFC1, to demonstrate FGFC1 as a new dual-target bioactive compound for thrombolytic therapy and to offer a promising evaluation method for the preclinical trials of thrombolytic drug candidates.

## Figures and Tables

**Figure 1 pharmaceuticals-17-01401-f001:**
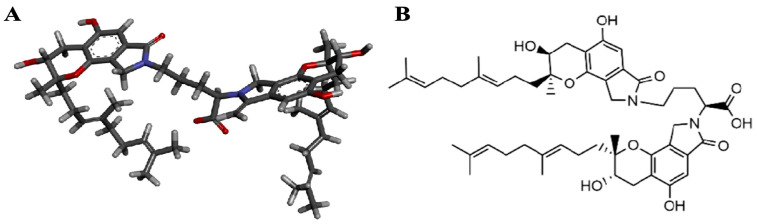
Chemical structure of FGFC1. Deep sea-derived isoindole alkaloid FGFC1 was isolated from secondary metabolites of *S*. *longispora* strain FG216, which was found to have fibrinolytic effects. Three-dimensional structure (**A**) and planar structure (**B**) of FGFC1 are shown.

**Figure 2 pharmaceuticals-17-01401-f002:**
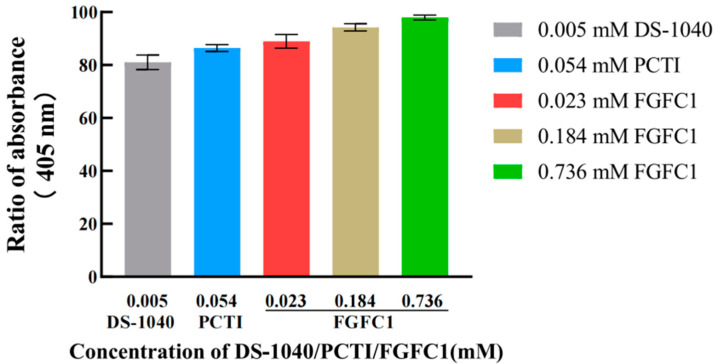
Inhibitory effects of DS-1040 (0.005 mM), PCTI (0.054 mM), and FGFC1 (0.023, 0.184, 0.368 mM) on TAFI activation. The TAFIa existence was detected by Hippuryl-Arg in the hPPP samples treated without (Blank) or with DS-1040 (0.005 mM), PCTI (0.054 mM), and FGFC1 solutions (0.023, 0.184, 0.368 mM), respectively. Absorbances at 405 nm relative to the Blank group are shown as the mean ± SD (n = 6).

**Figure 3 pharmaceuticals-17-01401-f003:**
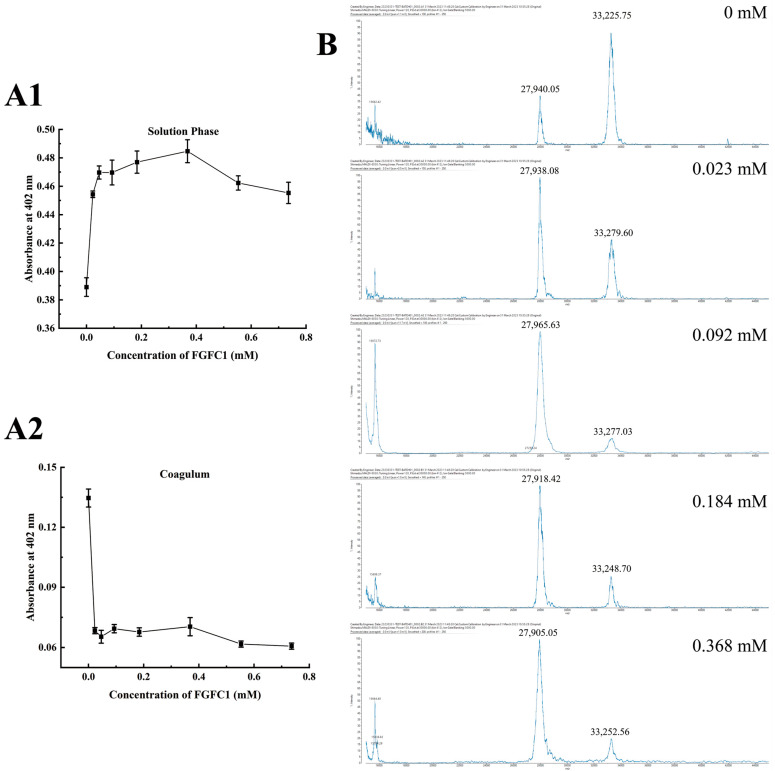
Regulatory effects of FGFC1 (0.023–0.368 mM) on TAFIa. (**A1**,**A2**) The TAFIa existence was detected by Hippuryl-Arg in the solution phase and the coagulum in the hPPP samples treated without (Blank) or with FGFC1 solutions (0.023–0.368 mM), respectively; (**B**) the 33 kDa/28 kDa monomers of TAFIa were detected by MALDI-MS in the solution phase in the hPPP samples treated without (Blank) or with FGFC1 solutions (0.023–0.368 mM), respectively. All values are indicated as the mean ± SD (n = 6).

**Figure 4 pharmaceuticals-17-01401-f004:**
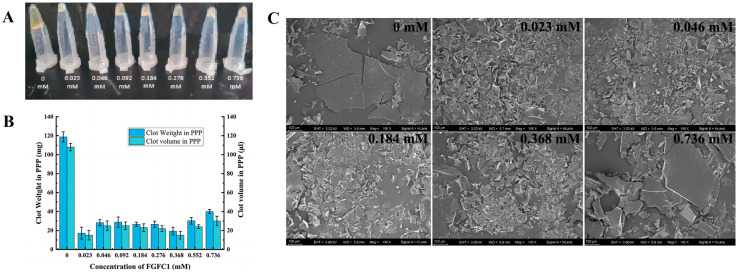
The inhibitory effect of FGFC1 (0.023–0.736 mM) on the coagulation level in the hPPP. (**A**,**B**) FGFC1 solutions (0.023–0.736 mM) suppressed the formation of the coagulum in the hPPP ex vivo, indicated by the volume and weight; all values are indicated as the mean ± SD (n = 6). (**C**) The inhibition on the coagulation level by the FGFC1 solutions (0.023–0.736 mM) observed by SEM at 500-fold, respectively. All the replicas were gold-coated and inspected under the electromagnetic/electrostatic composite lens. Each image is representative of 3 similar observed spots in 3 parallel experiments.

**Figure 5 pharmaceuticals-17-01401-f005:**
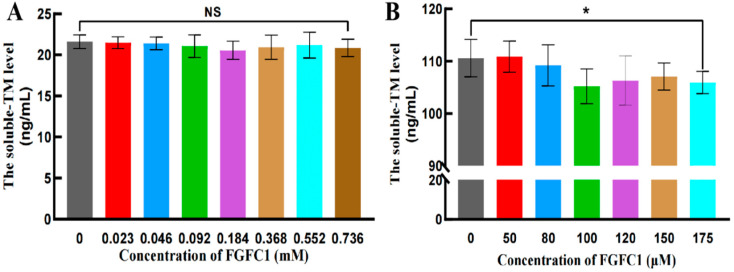
The soluble-TM levels in the hPPP and hCMEC/D3 cells. The soluble-TM levels were evaluated in the hPPP samples (**A**) and hCMEC/D3 cells (**B**), using ELISA. All values are indicated as the mean ± SD (n = 6). NS refers to no significant difference and an asterisk indicates a significant difference among various concentrations of FGFC1 solutions (* *p* < 0.025), determined by the Kruskal–Wallis test.

**Figure 6 pharmaceuticals-17-01401-f006:**
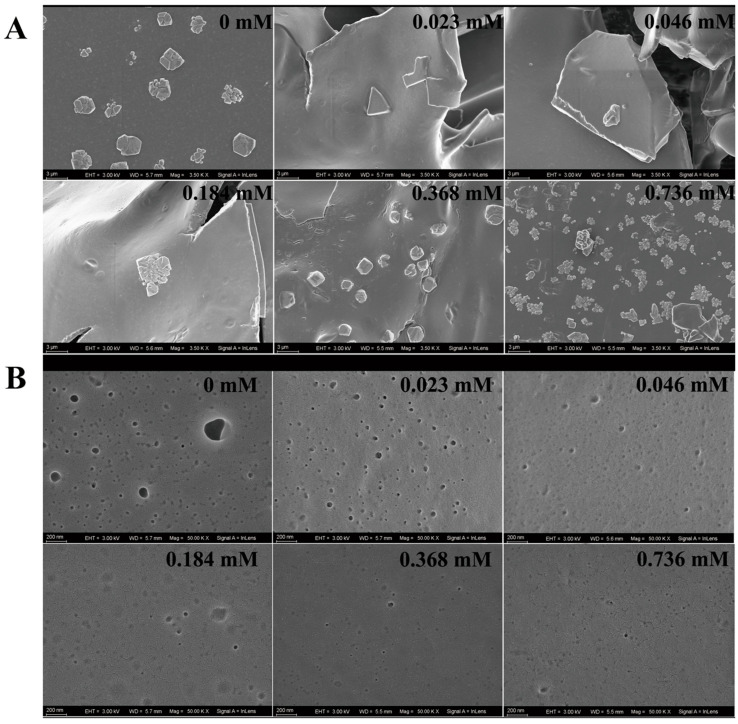
The influence of FGFC1 (0.023–0.736 mM) on the morphological characteristics of the coagulum in the hPPP. The inhibitions by the FGFC1 solutions (0.023–0.736 mM) on the clot protein crystallization level (**A**) and the PA anchoring (**B**) were observed by SEM at 3500- and 50,000-fold, respectively. All the replicas were gold-coated and inspected under the electromagnetic/electrostatic composite lens. Each image is representative of 3 similar observed spots in 3 parallel experiments.

**Figure 7 pharmaceuticals-17-01401-f007:**
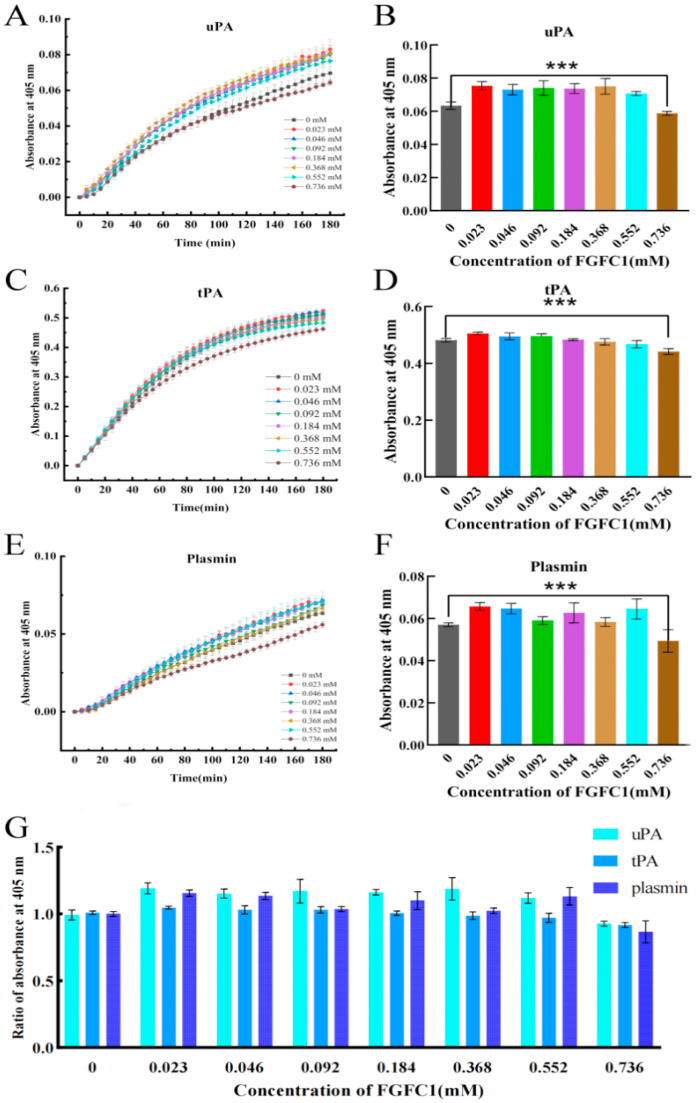
The TAFI-mediating fibrinolytic effect of FGFC1 in the hPPP. (**A**,**C**,**E**) The kinetic graph of FGFC1-influenced tPA, uPA, and plasmin activity in the hPPP samples. (**B**,**D**,**F**) FGFC1-influenced tPA, uPA, and plasmin activity and (**G**) the ratio of them in the hPPP samples after 150 min since the coagulum formed. All values were calibrated by the initial absorbance (405 nm) at 0 min in each group and indicated as the mean ± SD (n = 6). Asterisks indicate a significant difference among various concentrations of FGFC1 solutions (*** *p* < 0.001) determined by the Kruskal–Wallis test.

**Figure 8 pharmaceuticals-17-01401-f008:**
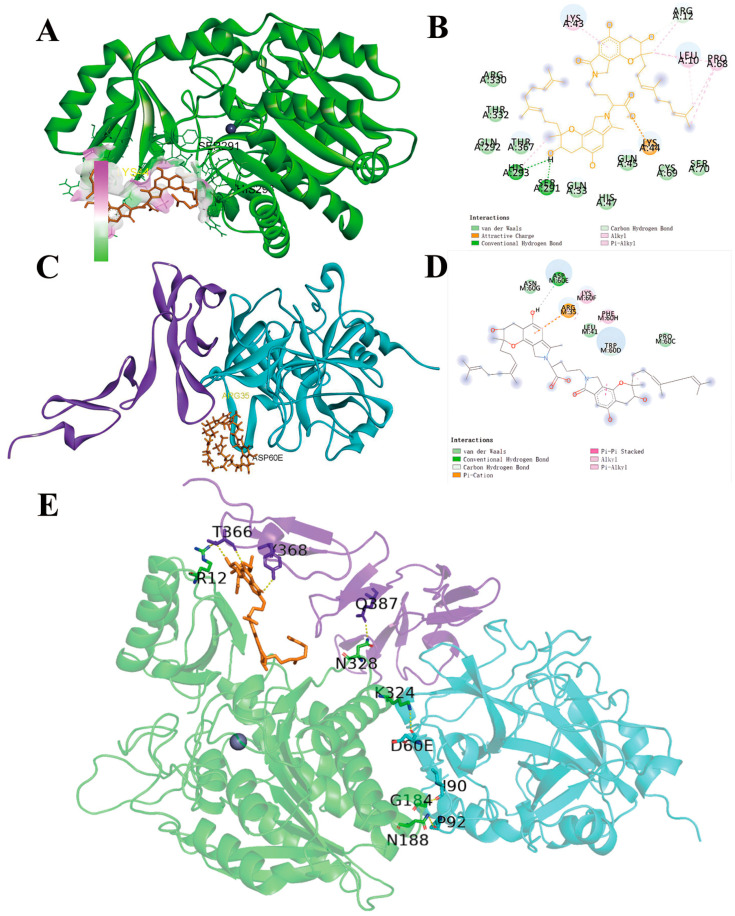
The schematic representation of the interaction between FGFC1 and the optimal conformation of TAFI. (**A**) The three-dimensional model of FGFC1-docking TAFI conformation (green: TAFI peptide, orange: FGFC1, gray: ZN^2+^); (**B**) the planar model of the binding sites on TAFI by FGFC1; (**C**) the three-dimensional model of FGFC1-docking TM conformation (blue: TM, thrombin peptide; orange: FGFC1; purple: the TME456 region of TM); (**D**) the planar model of the binding sites on TM by FGFC1; (**E**) the three-dimensional model of the TAFI-FGFC1-TM conformation.

**Figure 9 pharmaceuticals-17-01401-f009:**
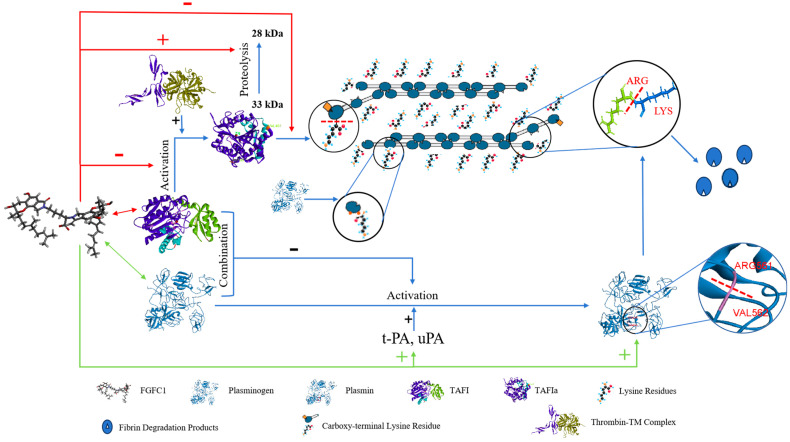
Pathway of TAFI-mediating profibrinolysis and anti-coagulation by FGFC1.

**Figure 10 pharmaceuticals-17-01401-f010:**
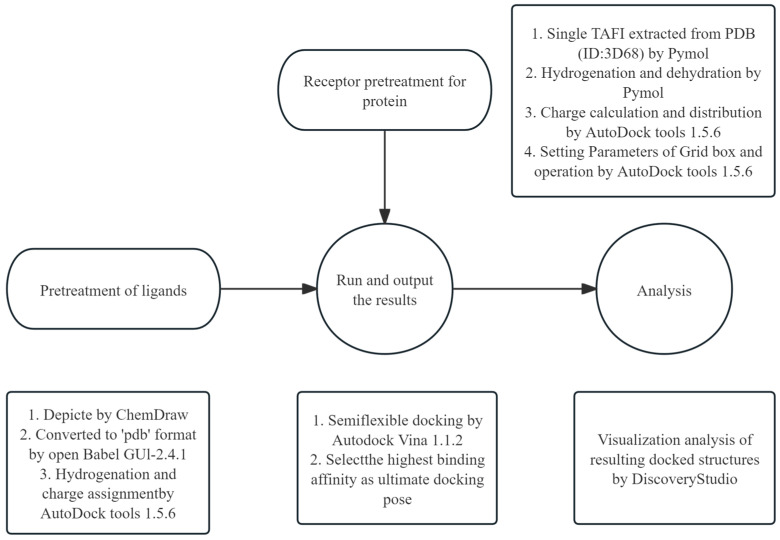
The molecular docking process. The Grid box interval was set at 126 × 126 × 126 (spacing of 0.503 Å) and the exhaustiveness parameter was set to 40. Note: The FGFC1 docking method allows for multiple distortions during sampling due to its flexible settings. However, this semi-flexible docking protocol has certain limitations.

**Table 1 pharmaceuticals-17-01401-t001:** Binding positions and energy in optimal conformation.

Compound Name	Binding Energy	Interaction		
		H Bond	Hydrophobic	Electrostatic
FGFC1-TAFI	−5.62 kcal/mol	SER291, HIS293	LEU10, ARG12, LYS43, PRO68	LYS44
FGFC1-TM	−0.52 kcal/mol	ASP60E	LYS60F, PHE60H, TRP60D	ARG35M
TAFI-FGFC1-TM		R12-T366, G184-I90, G188-P92, K324-D60E, N328-Q387		

**Table 2 pharmaceuticals-17-01401-t002:** MALDI-MS Data Acquisition Parameters.

Tuning	Linear
Polarity	Positive
Mass range	5–80 kDa
Laser rep. rate	200 Hz
Laser power	120
Accumulation rate (shots/profile)	50
Profiles	200
Pulsed extract	30,000.00
Ion gate blanking	5000.00

## Data Availability

The data presented in this study are available.

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
