# Peer review of "Novel Deep Sea Isoindole Alkaloid FGFC1 Exhibits Its Fibrinolytic Effects by Inhibiting Thrombin-Activatable Fibrinolysis Inhibitor"

_pharmaceuticals, 2024, doi:10.3390/ph17101401_

Round 1
Reviewer 1 Report
Comments and Suggestions for Authors
Title: A Novel Deep-sea isoindole alkaloid FGFC1 Exhibits its Fibri-nolytic Effects via Inhibiting Thrombin Activatable Fibrinolysis Inhibitor
Manuscript Number :
The manuscript entitled "A Novel Deep-sea isoindole alkaloid FGFC1 Exhibits its Fibri-nolytic Effects via Inhibiting Thrombin Activatable Fibrinolysis Inhibitor” is well written and the work plan is well organised. This manuscript explores the regulatory effects of a novel fibrinolytic compound, fungi fibrinolytic compound 1 (FGFC1), sourced from the deep sea, on TAFI. The efficacy of FGFC1 in inhibiting TAFI activation was compared to established TAFI inhibitors, DS-1040 and PCTI, in human platelet-poor plasma (hPPP). Molecular docking confirmed the interaction between FGFC1 and TAFI. FGFC1 exhibited concentration-dependent inhibition of TAFI activation, with significant effects observed at 0.023 mM, comparable to DS-1040 and PCTI. The study further demonstrated that FGFC1 inhibited TAFI-mediated coagulation (ex vivo and in vitro) by reducing clot aggregation, protein crystallisation, and platelet anchoring, as observed through scanning electron microscopy (SEM). Simultaneously, FGFC1 enhanced TAFI-mediated fibrinolysis, as supported by increased levels of tissue plasminogen activator (t-PA), urokinase-type plasminogen activator (u-PA), and plasmin. The effects of FGFC1 on TAFI were shown to be independent of TM-mediated control, indicating a distinct mechanism. FGFC1 was identified as a dual-target bioactive compound with both anti-coagulant and fibrinolytic properties. These findings highlight the critical role of TAFI in linking coagulation and fibrinolysis and suggest FGFC1 as a potential therapeutic agent for thromboembolic and hemorrhagic disorders associated with hematologic malignancies. Therefore, the manuscript can be accepted but only after providing suitable comments for the following queries.
Abstract
How do the effects of FGFC1 on TAFI activation and the coagulation/fibrinolysis balance vary across different cell types and tissue environments?
Introduction
What are the specific limitations and side effects associated with current thrombolytic drugs that make them less effective or safe for treating thromboembolic diseases?
What further research is needed to fully understand the mechanism of FGFC1’s TAFI-mediated thrombolytic effects, and how could this knowledge inform the development of more effective and safer thrombolytic therapies?
Comments on the Quality of English LanguageNeed to improve grammar in entire manuscript
Reviewer 2 Report
Comments and Suggestions for Authors
The work of Zhang et al analyzes the effects of FGFC1, a new fibrinolytic from deep sea, on TAFI. The authors demonstrate that 0.023 mM concentration provided suppression comparable to PCTI. The work is very interesting and well structured. I suggest the following integrations:
- More information on FGFC1 should be provided in the introduction;
- The results are well presented;
- A paragraph on future research perspectives would be useful with clear conclusions
Reviewer 3 Report
Comments and Suggestions for Authors
Dear authors, the manuscript entitled " A Novel Deep-sea isoindole alkaloid FGFC1 Exhibits its Fibrinolytic Effects via Inhibiting Thrombin Activatable Fibrinolysis Inhibitor" represents an interesting study in the field, showing the potential anti-coagulant properties of FGFC1, which is a a novel fibrinolytic compound 21 sourced from deep sea. However, major revisions are required before the manuscript further proceed.
1) In the introduction section, the figure 1 should be removed and placed as supplementary material, or totally be removed from the manuscript.
2) In the materials and methods section, paragraph 4.1. The Construction of hPPP-based ex-vivo Systemhe, the authors indicated that the hPPP employed in this study was isolated by the centrifugation of freshly col-293 lected whole blood from healthy volunteers. The authors need to include the total number of the volunteers enrolled in the study and also their age. Also, written informed consent according to the guidelines of Helsinki declaration for each patient should also be provided and sent to the mdpi, prior the manuscript further proceed.
3) In the materials and methods section, why the authors used the PPP instead of PRP or plasma for their experimental procedure. PPP is low in endogenous thrombin and poor in platelets so the coagulation to be performed requires further time. In my opinion, the same experipmental procedure should be used in PRP, plasma and whole blood to lead to safe conclusions.
4) The authors should describe in detail the production process of the hPPP, which used in this study.
5) In additon, the authors indicated that the FGFC1 in abstract, discussion and conclusion section, that FGFC1 can be used as therapeutic treatment for the patients with hematological issues. However, before the authors assume that FGFC1 may be used in humans, i propose the authors to use the FGFC1 in animal model to test the proposed hypothesis
Reviewer 4 Report
Comments and Suggestions for Authors
I sincerely appreciate the effort of the Authors to provide new knowledge about the drug that could be helpful in the patients with the life-threatening thromboembolic complications.
Content suggestions:
1. Why the Authors focused also on thromboembolic and bleeding events in the patients with malignancies – because of the limitations of thrombolysis in this indication ?
2. In this setting, is it usable also in a pregnancy or in further groups of patients, where thrombolysis is risky ?
Round 2
Reviewer 3 Report
Comments and Suggestions for Authors
Dear Authors,
You have succesfully addressed the majority of my comments. Well done!!